# A Histone Deacetylase (HDAC) Inhibitor with Pleiotropic In Vitro Anti-*Toxoplasma* and Anti-*Plasmodium* Activities Controls Acute and Chronic *Toxoplasma* Infection in Mice

**DOI:** 10.3390/ijms23063254

**Published:** 2022-03-17

**Authors:** Delphine Jublot, Pierre Cavaillès, Salima Kamche, Denise Francisco, Diana Fontinha, Miguel Prudêncio, Jean-Francois Guichou, Gilles Labesse, Denis Sereno, Corinne Loeuillet

**Affiliations:** 1CNRS UMR5525 TIMC-IMAG, Team BNI, Grenoble Alpes, CNRS, Grenoble INP, 38000 Grenoble, France; delphine.jublot@univ-grenoble-alpes.fr (D.J.); pierre.cavailles@univ-grenoble-alpes.fr (P.C.); salima.kamche@univ-grenoble-alpes.fr (S.K.); corinne.loeuillet@univ-grenoble-alpes.fr (C.L.); 2INSERM U1209, CNRS UMR 5309, Institute for Advanced Biosciences, Team ApicoLipid, Université Grenoble Alpes, 38000 Grenoble, France; 3CNRS UMR5525 TIMC-IMAG, Team TrEE, Grenoble Alpes, CNRS, Grenoble INP, 38185 Grenoble, France; 4Instituto de Medicina Molecular João Lobo Antunes, Faculdade de Medicina, Universidade de Lisboa, Avenida Professor Egas Moniz, 1649-028 Lisboa, Portugal; denise.francisco@hotmail.pt (D.F.); dfontinha@medicina.ulisboa.pt (D.F.); mprudencio@medicina.ulisboa.pt (M.P.); 5Centre de Biologie Structurale (CBS), INSERM, CNRS, Université de Montpellier, 34000 Montpellier, France; guichou@cbs.cnrs.fr (J.-F.G.); gilles.labesse@cbs.cnrs.fr (G.L.); 6Parasite Infectiology Research Group, Institut de Recherche Pour le Développement, Université de Montpellier, InterTryp, 34032 Montpellier, France; 7INSERM U1209, CNRS UMR 5309, Institute for Advanced Biosciences, Team Genetics Epigenetics and Therapies of Infertility, Université Grenoble Alpes, 38000 Grenoble, France

**Keywords:** anti-*Toxoplasma* drug, mice, IC_50_, HDACi, treatment, *Plasmodium*

## Abstract

Toxoplasmosis is a highly prevalent human disease, and virulent strains of this parasite emerge from wild biotopes. Here, we report on the potential of a histone deacetylase (HDAC) inhibitor we previously synthesized, named JF363, to act in vitro against a large panel of *Toxoplasma* strains, as well as against the liver and blood stages of *Plasmodium* parasites, the causative agents of malaria. In vivo administration of the drug significantly increases the survival of mice during the acute phase of infection by *T. gondii*, thus delaying its spreading. We further provide evidence of the compound’s efficiency in controlling the formation of cysts in the brain of *T. gondii*-infected mice. A convincing docking of the JF363 compound in the active site of the five annotated ME49 *T. gondii* HDACs was performed by extensive sequence–structure comparison modeling. The resulting complexes show a similar mode of binding in the five paralogous structures and a quite similar prediction of affinities in the micromolar range. Altogether, these results pave the way for further development of this compound to treat acute and chronic toxoplasmosis. It also shows promise for the future development of anti-*Plasmodium* therapeutic interventions.

## 1. Introduction

The phylum Apicomplexa includes parasitic genera of medical and veterinary importance, such as *Plasmodium*, *Toxoplasma*, and *Babesia*, the causative agents of malaria, toxoplasmosis, and babesiosis, respectively.

According to the World Health Organization (WHO), in 2020, there were an estimated 241 million cases of malaria, resulting in 627,000 deaths [1]. Five species belonging to the *Plasmodium* genus cause malaria in humans: *P. falciparum*, *P. vivax*, *P. ovale*, *P. malariae*, and, in parts of southeast Asia, the monkey malaria parasite, *P. knowlesi*. Almost all deaths are caused by *P. falciparum* [2]. In addition, the intensive use of the available treatments, participating at least in part in the emergence of multidrug resistance, contributes to a higher risk of transmission, morbidity, and mortality [3].

Toxoplasmosis is an opportunistic infectious disease infecting one-third of the population worldwide [4]. Sixteen haplogroups belonging to six ancestral groups (clades) are described [5]. Type II parasites are responsible for human toxoplasmosis in Europe and North America. These strains are associated with asymptomatic illness in immunocompetent individuals [6]. However, toxoplasmosis can be severe in immunocompromised individuals or neonates. In addition, less frequent “exotic” genotypes have been recorded, especially in South America, some of which are responsible for severe disseminated toxoplasmosis in immunocompetent individuals with a possibly lethal outcome [7,8,9,10]. A possible association between toxoplasmosis and behavioral alteration is supported by experimental animal infection [11,12], and an association between the presence of anti-*Toxoplasma* antibodies and schizophrenia, bipolar disorder, or obsessive–compulsive disorder has been reported [13,14].

Combinations of anti-*Toxoplasma* drugs, pyrimethamine–sulfadoxine (Fansidar or Malocide–Adiazine), are widely used against the acute phase of infection. These therapies are efficient against tachyzoites, the proliferative stage of the parasite. Nevertheless, in the absence of a long-term treatment follow-up, relapse rates of up to 80% are recorded [15], and parasite resistance to these treatments has also been reported [16]. In addition, renal and liver damage, diarrhea, and blood cell deficiency are significant side effects of treatment [17,18]. Currently, no alternative is available to eliminate the encysted form of the parasite, which can be reactivated in immunocompromised individuals, such as human immunodeficiency virus (HIV)-infected patients and recipients of organ transplants undergoing immunosuppressive treatment, causing encephalitis, severe eye damage, and, in some cases, death [19,20,21]. Overall, there is a need for additional antiparasitic molecules that consider the diversity of the clinical aspects of toxoplasmosis and the side effects of existing treatments. The challenge in designing new anti-*Toxoplasma* drugs includes the development of molecules with activity against the tachyzoite and bradyzoite forms of the parasite that can cross the placental and blood–brain barriers. They must also be nontoxic and affordable [22]. The main characteristics of the ideal molecule to fight toxoplasmosis, as reviewed by McFarland [23], include an IC_50_ below 10 μM, a high therapeutic index, and an increase in the survival time of infected hosts when challenged in vivo with a lethal parasite dose.

Several parasite forms with specific proliferative capacity are present within an infected host, including tachyzoites (highly proliferative), bradyzoites (less proliferative, tissue cyst form), and sporozoites (oocysts form) [6]. The passage from the bradyzoite to the tachyzoite state is controlled by epigenetic mechanisms involving histone deacetylases (HDACs) [24]. An increasing number of compounds potentially inhibiting histone deacetylases (HDACs) revealed high anti-*T. gondii* and/or anti-*Plasmodium* activity in vitro or animal models [25,26,27,28,29,30,31]. Therefore, these enzymes are relevant targets for treating cysts.

By screening a series of aminophenylhydroxamates and aminobenzylhydroxamates for their antiparasitic activities, we characterize a molecule named JF363, with high selectivity and HDAC6 inhibitory activity, which correlated to its capacity to inhibit type I and type II *T. gondii* proliferation in vitro [25,26]. Here, we have analyzed this compound’s in vitro antiproliferative capacity against other *Toxoplasma* genotypes and have further assessed its in vitro activity against *P. berghei* and *P. falciparum*, hepatic and blood stages, respectively. Finally, using an experimental model of infection, we ascertained compound JF363’s ability to circumvent toxoplasma parasite dissemination during the acute phase of the disease and act on the chronic stage of the illness. Besides, the biodistribution of the drug was assessed in noninfected and infected mice.

## 2. Results

### 2.1. Anti-Proliferative Activity of JF363

#### 2.1.1. Pleiotropic In Vitro Anti-*Toxoplasma* Activity

To investigate the effect of the JF363 on parasite proliferation, human HFFs were infected and treated with JF363 (concentrations ranging from 0 to 2 μM). After 24 h of culture, parasites were enumerated within the parasitophorous vacuole, and the IC_50_ was calculated for each parasite strain (Table 1). We confirmed the high efficiency of JF363 to inhibit the proliferation of *T. gondii* strains belonging to the haplogroup I and haplogroup II parasites with IC_50_ values of 0.56 and 0.27 μM, respectively. For other strains, the IC_50_ values ranged between 0.17 and 0.43 μM.

#### 2.1.2. Anti-*Plasmodium* Activity

The activity of JF363 against the obligatory and clinically silent hepatic stage of the rodent malaria *P. berghei* (Pb) and the blood stage of the human malaria *P. falciparum* (Pf) parasites was assessed in vitro.

The compound displayed nanomolar-range dose-dependent activity against Pb hepatic infection (Figure 1), with an IC_50_ of 98.9 ± 22.7 nM. Furthermore, JF363 did not present toxicity towards the hepatic host cells at the tested concentrations, as shown by the similar cell confluency values observed for compound- and vehicle-treated cells.

Having assessed the in vitro activity of the compound against the hepatic stage of *Plasmodium* infection, the in vitro impact on the blood stages of the human *P. falciparum* parasite was evaluated (Figure 2). As assessed by flow cytometry, the compound showed an inhibitory effect on the parasite growth at concentrations ranging from 200 to 2000 nM. The drug’s IC_50_ was determined in three independent experiments and was 230.0 ± 15.0 nM.

### 2.2. In Vivo Anti-Toxoplasma Activity

#### 2.2.1. Effect of JF363 on Mouse Survival during the Acute Phase of *T. gondii* Infection

To determine the in vivo anti-*Toxoplasma* effect of the molecule JF363 during the acute phase of infection, we analyzed the survival rate of non-inbred CD1 mice infected by the type I RH strain treated for 20 days. All healthy mice treated with drugs survived (data not shown), and they did not present any sign of body hair loss, erythema, or organ failure. This indicates that the JF363 molecule is not toxic. All untreated and infected mice succumbed by day 9 (Figure 3). After seven days of continuous oral administration, we observed a significant dose-dependent effect (*p* < 0.0001), as demonstrated by an increase in mouse survival by day 11 and day 14 for mice treated with 40 or 160 mg/kg. Thus, molecule JF363 used alone delays mouse death significantly.

#### 2.2.2. Effect of the JF363 on Cyst Formation in Chronically Infected Mice

To assess the efficacy of compound JF363 on *Toxoplasma* cyst formation, infected CBA mice were treated for 15 days, two months post-infection. Then, 15 days later, mice were sacrificed, and brain cysts were counted. Many cysts (168 ± 95, range 76 to 312) were detected in the infected nontreated mice, whereas in infected mice treated with the compound JF363 (40 mg/kg), a significantly (*p* < 0.05) lower number of cysts (57 ± 29; range 21–85) was detected (Figure 4). Thus, the JF363 significantly decreases the number of cysts in the brain of infected mice.

### 2.3. Bioavailability

A substantial in vivo deionization of the radioactivity is recorded in the thyroid, salivary glands, stomach, and intestine following the oral gavage of mice with the radiotracer. This deionization has already been reported [32]. An example of the sagittal view gathered during the experiment is given (Figure 5). The iodinated JF363 elimination occurred primarily through the urinary tract. Neck ganglia were visible on SPECT imaging at 30 min pi and 3.5 h pi in mouse #2 (8.5% and 9.5% ID/cm^3^, respectively) and to a lesser extent in mouse #3 (4% and 5% ID/cm^3^, respectively), and this activity decreased to ~1% at 24 h pi in both mice (Table 2).

Little radioactivity was recorded in the brain, with no particular accumulation detected in the brain slides. Nevertheless, the radioactivity detected was 3, 9, and 2.6 times more intense in the brain (revealing the presence of our drug JF363) when looking for the ratios of infected/control mice for the organ/blood. This suggests a permeability of the blood–brain barrier to the JF363 molecule (Figure 6).

When considering the ratio of infected/control mice for the organ/blood radioactivity, we noticed an increase in the presence of JF363 in almost all the organs of the infected mice except for the small intestine, pancreas, spleen, and kidney. Nevertheless, a modification of the molecule diffusion cannot be ruled out following its iodination.

### 2.4. In Silico Modeling and Docking

The five annotated HDACs in ME49 were modeled by extensive sequence–structure comparisons with HDACs whose structures have been solved experimentally. Good to excellent models could be obtained thanks to good sequence coverage (85–100% for the conserved catalytic domain) and good to high sequence identity (30–65%). Comparative docking of ligands co-crystallized with the structure templates confirmed proper modeling of the binding site. Although the catalytic zinc could not be used during the docking in @TOME, convincing docking of JF363 was obtained in the active site of the five modeled HDACs, thanks to the use of shape restraints deduced from the co-crystallized ligands. The resulting complexes showed a similar binding mode in the five paralogous structures for the metal-chelating phenyl-hydroxamate moiety, while the methoxybenzyl moiety shows two opposite orientations at the entrance of the active site (see Figure 7). This screening predicts quite similar affinities (differences within the range of error) in the micromolar range (in the absence of the favorable contribution of the metal center).

## 3. Discussion

We previously highlighted the potential of an HDAC inhibitor (JF363) to efficiently control the intracellular proliferation of some type I and type II *T. gondii* strains [25]. Here, we confirmed the antiparasitic potential of this compound by assessing its capacity to act against a larger panel of *Toxoplasma* strains and the hepatic and blood stages of *Plasmodium* infection. In addition, we investigated the biodistribution of the compound Jf363 after oral delivery. Finally, we probed its efficiency in controlling *Toxoplasma* infection during the acute phase and the formation of brain cysts during the chronic phase.

In vitro, in an HFF model of infection, the IC_50_ of JF363 against *T. gondii* varied from 0.17 to 0.56 μM, depending on the haplogroup. Furthermore, the IC_50_ recorded for the 363 compounds fell in the range of activity of the primary drug used to treat toxoplasmosis, pyrimethamine (IC_50_ of 0.4 μM) [33]. These data show the potential of JF363 to be used against *Toxoplasma* with a wide range of genotypes, even the hypervirulent ones or those from the wild biotope such as the GUY-JAG1 strain [34]. This is of interest since humans will be more and more in contact with such strains coming from the wild biotope, especially in South America, as demonstrated by the increasingly diverse genotypes reported from congenital toxoplasmosis in South America [35]. In addition, Brazilian *Toxoplasma* strains would require higher pyrimethamine or sulfadiazine dosages, and sulfadiazine-resistant isolates were identified [36,37,38].

The drug also displayed IC_50_ values below the micromolecular concentration range for both the hepatic (~0.1 μM) and the blood (0.23 μM) stages of *Plasmodium* infection. This dual-stage activity of the compound warrants its further exploration as an effective anti-*Plasmodium* agent. Furthermore, the absence of detected specificities against modeled Tg HDACs matches well with the experimentally observed pleiotropic effect of the compound JF363 against the various *T. gondii* we tested. Further structural analyses would help in the development of more potent and more specific inhibitors. Such a pleiotropic antiparasitic drug that is potent against parasites belonging to the apicomplexan phylum reinforces the interest in using this molecule to treat infections caused by these pathogens.

Animal models have been extensively used to describe infection pathology and identify new effective drugs for treating toxoplasmosis. Most laboratory animal models are susceptible to *Toxoplasma* infection, with the resulting outcome depending on the strain. The infection can be acute, subacute, or chronic and be easily monitored by the survival of animals or by histopathological examinations. In animals, acute infections with highly virulent strains induce disseminated infection with significant pulmonary and brain involvement. They thus can be used to assess the efficacy of drugs in these tissues [39]. Oral delivery of the compound JF363 significantly increases the survival rate during the acute phase of the infection (challenge by a lethal dose of RH *T. gondii* strain). The JF363 drug increased mouse survival by two days at the concentration of 40 mg/kg and eight days at 160 mg/kg. In our experiment, animals were treated with the drug once a day, beginning 6 h after RH infection and then for 7 days. Treating animals twice a day might increase their survival or rescue them, as is the case for a cyclic GMP-dependent protein kinase inhibitor [40]. In addition, increasing the drug dose to 200 or 250 mg/kg, in the range of dose used for niclosamide [41], would improve the survival of mice and perhaps ultimately rescue them.

To test the drug effect on cyst reduction, mice were inoculated with type II Prugniaud tachyzoites and treated two months after infection, a time at which we can consider that the cyst burden is relatively stable. We noticed that the 40 mg/kg dose effectively diminishes cyst burden in the brain by 65% compared to untreated animals. The majority of other drugs presenting an effective reduction in brain cyst number have been tested in combination, at a higher dose, and for a more extended period, more than two weeks [42]. Here, the JF363 drug demonstrates a fairly good anticyst activity when used alone and in a small amount. This is not the first time that HDAC inhibitors have been reported to hold promise as a new antiparasitic drug. Maubon et al. reported that the FR235222 drug is efficient in vitro, acting on parasite proliferation and ex vivo cyst formation. Still, its in vivo efficiency remains to be investigated [29].

We recorded an increase in drug presence in almost all tested organs in infected mice. However, more importantly, the concentration was elevated in most organs targeted by parasites, organs such as the brain, heart, muscle, and lung. Considering the high drug efficiency we recorded on tachyzoites in vitro, and knowing that the radioactivity recorded even if present was lower than that in other tested organs, the drug concentration seems sufficient to act on bradyzoites, explaining the observed reduction in brain cyst burden. The detection of the iodinated JF363 compound in the abdominal fat can be due to the carrier we used in this study, corn oil, which favors the drug accumulation in fatty tissues [43].

## 4. Materials and Methods

### 4.1. Parasites

B. Striepen (Athens, Greece) kindly provided RH-YFP (haplogroup 1, clade A) and Prugniaud (haplogroup 2, clade D). D.J. Bzik (Lebanon, USA) kindly provided PruDeltaKu80:HX strain. The VEG (haplogroup 3, clade C), MAS (haplogroup 4, clade B), GUY008-ABE (haplogroup 5, clade F), GUY021-TOJ and GUY009-AKO (haplogroup 10, clade F), and GUY-JAG1 (haplogroup 11, no clade) strains were purchased from the Centre de Ressources Biologiques (CRB) *Toxoplasma*, Reims, France.

Tachyzoites of *T. gondii* strains were maintained under standard procedures by serial passage onto human foreskin fibroblast monolayers (HFFs) in D10 medium (DMEM supplemented with 10% heat-inactivated fetal bovine serum, 1 mM glutamine, 500 units/mL of penicillin, and 50 mg/mL of streptomycin) at 37 °C in a humidified atmosphere containing 5% CO_2_. The parasites were collected before the experiment, centrifuged at 500× *g* for 7 min, suspended in D10 medium, and counted.

*Plasmodium berghei* sporozoites expressing luciferase [44] were obtained by dissection of salivary glands of infected female *Anopheles stephensi* mosquitoes, reared at Instituto de Medicina Molecular (Lisbon, Portugal). In addition, the chloroquine-sensitive NF54 strain (PfNF54) strain was cultured according to standard protocol [45].

### 4.2. Cell Culture and In Vitro Antimicrobial Test

#### 4.2.1. In Vitro Anti-*Toxoplasma* Activity

To assess the drug activity on *Toxoplasma gondii,* confluent HFF monolayers were infected with parasites, 5 × 10^4^ per well, then centrifuged for 30 s at 500× *g* and incubated for 30 min in a water bath at 37 °C to allow invasion. The wells were then washed three times with phosphate-buffered saline (PBS) to eliminate extracellular parasites, and drugs were added at various concentrations ranging from 1 to 2 μM in triplicate. After 24 h incubation at 37 °C in a humidified atmosphere containing 5% CO_2_, Hoechst 33342 (5 μg/mL) was added to host cells infected by parasites and further incubated for 20 min. After two washes, cells were fixed with 3.7% formaldehyde for 10 min at 37 °C.

In experiments using nonfluorescent parasites, *Toxoplasma* vacuoles were fixed for 15 min with 3.7% formaldehyde and stained with anti-GRA1 mAb diluted 1/500 in 2% bovine serum albumin (BSA) and 0.1% Triton X-100 in PBS, for 1 h at room temperature. After two washes, goat serum anti-mouse IgG (H + L)–Alexa-488 (1/500) was added for 20 min. Finally, Hoechst 33342 was added as described above.

The number of infected cells, i.e., cells harboring parasitophorous vacuoles, and the number of parasites per vacuole were determined using an Olympus ScanR microscope (×20 objective) and ScanR software. First, the parasitic index (PI) in % compared to the control was calculated as follows: PI = ((number of parasite/100 cells in treated wells) × (% of infected HFFs in treated well)/(number of parasite/100 cells in untreated wells) × (% of infected HFFs in untreated wells)) × 100. Then, the 50% inhibitory concentration (IC_50_) was calculated using Prism software (Prism 4 for Mac OS X, version 5.0b, San Diego, CA, USA).

Huh7 cells, a human hepatic cell line employed in the assessment of drug activity against the hepatic stage of *P. berghei* infection, were cultured in RPMI 1640 supplemented with 10% (*v*/*v*) fetal bovine serum, 1% (*v*/*v*) penicillin/streptomycin, 1% (*v*/*v*) glutamine, 1% (*v*/*v*) nonessential amino acids, and 10 mM N-2-hydroxyethylpiperazine-N′-2-ethanesulfonic acid (HEPES) at 37 °C and 5% CO_2_. For infection, Huh7 cells were seeded in 96-well plates at 1 × 10^4^ cells/well.

#### 4.2.2. In Vitro Anti-*Plasmodium* Activity

Liver stage. The action of JF363 against the liver stage of *P. berghei* infection was assessed by bioluminescence, as previously described [46]. Briefly, one day before infection, Huh7 cells were seeded in 96-well plates. JF363 stock solution was prepared in dimethyl sulfoxide (DMSO) and serially diluted in infection medium (culture medium supplemented with 50 µg/mL of gentamicin and 0.8 µg/mL of fungizone) to obtain the test concentrations (1, 10, 25, 50, 10, 250, 500 nM). These were added to the cells and incubated for 1 h at 37 °C and 5% CO_2_, after which 1 × 10^4^ sporozoites were added to the wells. Plates were centrifuged at 1800× *g* for 5 min and incubated at 37 °C with 5% CO_2_ for the assay duration. At 46 h post-infection, the impact of the compound on cell viability was assessed by the AlamarBlue assay (Invitrogen, Carlsbad, CA, USA), following the manufacturer’s instructions. Following the manufacturer’s instructions, the drug’s effect on the parasite load was next evaluated using the Firefly Luciferase Assay Kit 2.0 (Biotium, Fremont, CA, USA). For IC_50_ determination, nonlinear regression analysis was employed to fit the normalized results of the dose–response curves, using GraphPad Prism 8 (Prism 4 for Mac OS X, version 5.0b, San Diego, CA, USA).

Blood stage. Ring-stage synchronized cultures of PfNF54 at 2.5% hematocrit and at approximately 1% parasitemia were incubated with drugs or DMSO (vehicle control) in 96-well plates, for 48 h, at 37 °C in a 5% CO_2_ and 5% O_2_ atmosphere. The stock solution of JF363 was prepared in DMSO. Working solutions (1, 10, 100, 200, 500, 1000, 2000 nM) were prepared from the stock solution in a complete malaria culture medium (CMCM), which consisted of RPMI 1640 supplemented with 25 mM HEPES, 2.4 mM L-glutamine, 50 μg/mL gentamicin, 0.5% *w/v* AlbuMAX (Gibco BRL), 11 mM glucose, 1.47 mM hypoxanthine, and 37.3 mM NaHCO_3_. Five microliters of the culture (approximately 800,000 cells) was stained with the DNA-specific dye SYBR green I for each measurement. Following 20 min incubation in the dark, the stained sample was analyzed by flow cytometry. For each flow cytometry measurement, approximately 100,000 events were analyzed. All samples were analyzed in triplicate, and three different experiments were performed and analyzed using GraphPad Prism 8 (Prism 4 for Mac OS X, version 5.0b, San Diego, CA, USA).

### 4.3. Acute Phase Study

Non-inbred CD1a mice (Janvier Labs, Le Genest-Saint-Isle, France) were intraperitoneally infected with 100 tachyzoites of RH YFP strain in 200 mL of PBS. Six hours post-infection and then once a day for 7 days, mice were treated *per os* with either 40 or 160 mg/kg of JF363 prepared in corn oil/5% DMSO. Control animals were treated with the vehicle alone. Animals were monitored each day and sacrificed when the limit points were reached.

### 4.4. Chronic Phase Study

Inbred CBA mice (Janvier Labs, Le Genest-Saint-Isle, France) were infected with 500 tachyzoites of Pru-DeltaKu80::HX strain. One month post-infection, effective infection was assessed by ELISA as described previously [47]. Two months post-infection, mice were treated with 20 or 40 mg/kg of JF363 (corn oil/5% DMSO) for 15 days. Animals were sacrificed 15 days later, corresponding to 3 months post-infection, and brain cysts were counted. After removal and homogenization of mouse’s brain in 4 mL of PBS, suspensions were clarified by gentle incubation in proteinase K buffer (proteinase K 0.4 mg/mL, 10 mM Tris pH 8, ethylenediaminetetraacetic acid (EDTA) 1 mM, sodium dodecyl sulfate (SDS) 0.2%, sodium chloride 40 mM) for 15 min at 56 °C. The reaction was stopped by adding phenylmethylsulfonyl fluoride (PMSF) 2 mM and further incubation for 5 min at room temperature. The suspension was then washed with PBS and resuspended with fluorescein isothiocyanate (FITC)–Dolichos biflorus agglutinin (Vector Laboratories, CA, USA) 20 mg/mL for 30 min, at room temperature. At the end of the incubation, the suspension was washed with PBS and resuspended in a final volume of 1 mL of PBS distributed into six-well culture plates (1 brain per well). The number of cysts was counted by exanimation under an Axiovert 40 CFL inverted fluorescence microscope, Zeiss [48].

### 4.5. In Vivo Biodisponibility

Chemical structures of the compounds JF363 and iodo-JF363 are given in Figure 8. The synthesis of the JF363 was performed as previously described [25].

#### 4.5.1. Synthesis of the Iodo-JF363 Compound

The synthetic scheme of the iodo-JF363 is given in the Appendix A. Briefly, 4-aminobenzoate ethyl (1 eq.; 1.35 mmol) was dissolved in dry dichloromethane (DCM) (0.2 M). Diisopropylethylamine (DIEA) (6 eq.), hydroxybenzotriazole (HOBt) (1 eq.), and 4-iodo-3-methoxybenzeneacetic acid (1 eq.) were added successively at room temperature. The reaction mixture was concentrated following the completeness of the reaction and probed with thin liquid layer chromatography analysis (TLC). The residue was then taken up with ethyl ethanoate (EtOAc). The organic phase was washed with a solution of 1 M HCl, a saturated solution of NaHCO_3_ and brine, dried over Na_2_SO_4_, filtrated, and concentrated. Flash chromatography using a mixture of ethyl acetate and hexane was performed on the crude product purified to afford the amide (123 mg, yield 34%). Amide derivative (1 eq.; 0.28 mmol) was dissolved in H_2_O/MeOH 1/1 (4 mL). KOH (2 eq.) was added, and the reaction mixture was heated at 40 °C for 45 min. The reaction mixture was taken up with 20 mL of water, and the aqueous phase was washed with 20 mL of EtOAc; then, the aqueous phase was acidified to pH 2 with a solution of 1 M HCl. The aqueous phase was extracted three times with 20 mL of EtOAc. The organic phases were combined and dried over Na_2_SO_4_, filtrated, and concentrated to afford the acid derivative (110 mg, yield 99%). Acid derivative (1 eq.; 110 mg) was dissolved in dry dimethylformamide (DMF) (5 mL). Ethyl chloroformate (1.2 eq.) and N-methyl-morpholine (1.3 eq.) were added successively at 0 °C. After 10 min, a hydroxylamine (2 eq.) solution in methanol (MeOH) (10 mL) was added, and the reaction mixture was warmed up to room temperature. After 15 h, the reaction mixture was concentrated. The residue was taken up with EtOAc, and the organic phase was washed with a saturated solution of NaHCO_3_ and brine, dried over Na_2_SO_4_, filtrated, and concentrated. The crude product was purified by flash chromatography using reverse phase to afford the hydroxamate derivative iodo-JF363 (48 mg; yield 38%). The characterization and purification data of compound iodo-JF363 are given as Appendix A.

#### 4.5.2. Radiolabeling with Iodine-125

An isotopic exchange was performed to obtain the radiolabeled iodinated JF363. Briefly, 2 mg of the iodinated molecule was incubated in acetone with 3 mg of cold Na127I and 55.5 MBq of radioactive Na125I (PerkinElmer) for 1 h at 95 °C in a heat block. After evaporation of acetone, the radiotracer was solubilized in corn oil; 5% DMSO was also added to mimic the previous conditions. Quality controls were performed by TLC. Radioactive purity for all radiolabeling was found to be >98%, and the stability of the radiolabeled JF363 in the medium was good up to 72 h.

#### 4.5.3. SPECT/CT Imaging and Ex Vivo Biodistribution in Mice

Three female CD1 mice (31.7 ± 1.3 g; Janvier Labs, Le Genest-Saint-Isle, France) and two female mice infected with *T. gondii* obtained as previously described (19.5 ± 2.1 g) were studied. The previous study gave the radiotracer by oral gavage (administered dose: 0.73 ± 0.25 MBq/g). Before imaging, mice were anesthetized using isoflurane (2% in a mixture of air: O_2_) and positioned in a thermostatic bed. Whole-body single-photon emission computerized tomography/computed tomography (SPECT/CT) acquisitions were performed 0.5, 3.5, and 24 h after injection using a dedicated system (nanoSPECT-CT; Mediso, Budapest, Hungary). First, CT and SPECT acquisitions were reconstructed and fused using Nucline software (Mediso, Budapest, Hungary). This fusion allowed drawing a volume of interest (VOI in cm^3^) on selected/visible organs, and SPECT quantification based on CT was then performed using VivoQuant (InviCRO, Boston, MA, USA). Results were expressed in %ID/cm^3^ or %ID/g (percentage of injected dose per cubic centimeter or gram of tissue).

Twenty-four hours after injection and immediately following SPECT/CT image acquisitions, anesthetized mice were euthanized using CO_2_, and tissue samples (brain, heart, stomach, liver, salivary glands, abdominal fat, small intestine, muscle, pancreas, lungs, spleen, kidneys, blood, thyroid, large intestine, genital tract) and urine were harvested. Samples were weighed, and their radioactivity was determined with a gamma counter (Wizard2, PerkinElmer, Courtaboeuf, France). Results were corrected for decay, injected dose, and organ weight and expressed as % ID/g because the bodyweight (BW) of mice was different between both groups (normal and infected).

Standard uptake values (SUVs) were also calculated ((tissue activity/gram of tissue)/ID × BW). Due to high activity detected in the animal blood, results were also expressed as a ratio between %ID in a gram of organ and %ID in a gram of blood (org/blood ratio). Then, to assess the effect of *Toxoplasma* infection on the biodistribution of the iodoJF363, the ratios of radioactivity (infected/noninfected) were calculated.

Because *T. gondii* can form cysts preferentially in the brain, 20 µm thick brain slices, together with reference organs, were exposed for several days on an autoradiographic film which was then scanned using a phosphoimager (Fujifilm BAS-5000, FUJIFILM, Montigny, France). Slices were then stained with hematoxylin–eosin to detect cysts, thereby quantifying the radiotracer uptake (% ID/g).

### 4.6. Structure Modeling and Ligand Docking

Sequence modeling and ligand docking were performed on HDACs from *T. gondii* strain ME49 using the server @TOME as previously described [25]. However, the sequence of one truncated HDAC (Uniprot: S8GBW3_TOXGM) was complemented after TBLASTn searches to recover the missing part in its C-terminus. As a result, the new sequence better matched the full-length sequence of its orthologue in *T. gondii* strain VEG and allowed the modeling of a complete and stable catalytic domain. In addition, sequence–structure alignment and modeling allowed us to gather templates for focused ligand docking of the molecule JF363. The results are made available at http://atome.cbs.cnrs.fr/HDAC_Tgondii_ME49_JF363, accessed on 22 February 2022.

## 5. Conclusions

In conclusion, we demonstrated that compound JF363 is active against several *Toxoplasma* strains with different genotypes, delaying mouse death during the acute phase of infection and significantly reducing the burden of brain cysts. In addition, this drug has a high in vitro selectivity index [25], is relatively small, and can be easily synthesized. Furthermore, JF363 also displayed in vitro activity in the nanomolar range against the hepatic and the blood stages of *P. berghei* and *P. falciparum*, respectively. Moreover, it is not toxic in vitro and in vivo and can be delivered orally. Thus, our results warrant further exploration of JF363 as a pharmacological tool to fight human toxoplasmosis and malaria.

## Figures and Tables

**Figure 1 ijms-23-03254-f001:**
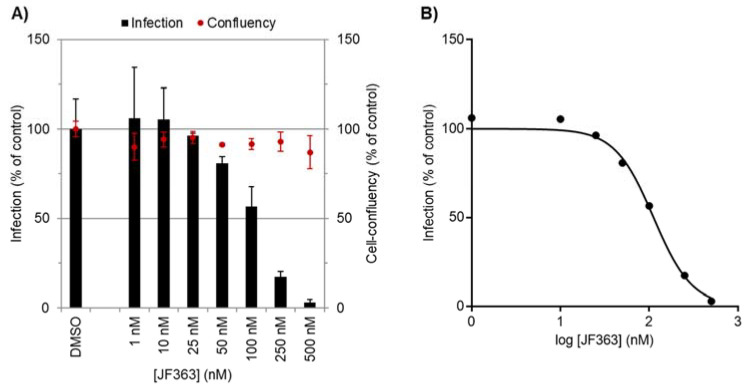
The dose-dependent activity of JF363 against *P. berghei* hepatic infection in vitro. (**A**) Total parasite load (infection scale, bars) and cell viability (cell confluency, dots). Results were normalized to the negative control, the drug vehicle DMSO, and are represented as mean ± SD. (**B**) Dose–response curve. A representative experiment out of independent 3 biological replicates.

**Figure 2 ijms-23-03254-f002:**
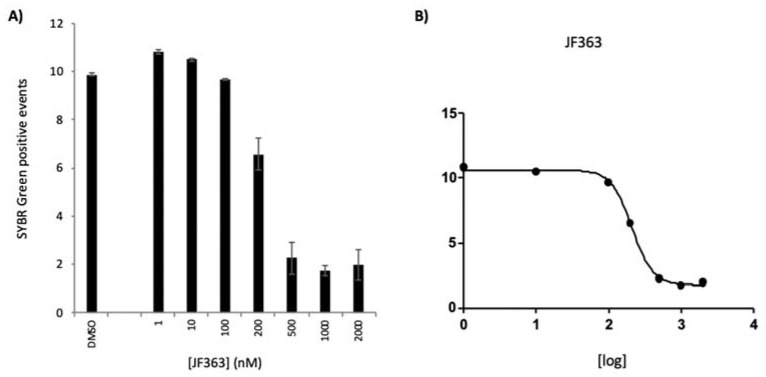
The dose-dependent activity of JF363 against *P. falciparum* erythrocytic infection in vitro. (**A**) Percentage of *P. falciparum*-infected red blood cells in the presence of different concentrations of JF363 or the drug vehicle, DMSO. A ring stage synchronized culture (~1% parasitemia) of *P. falciparum* NF54 was incubated for 48 h with increasing concentrations of JF363 (1, 10, 100, 200, 500, 1000, and 2000 nM), and a percentage of DMSO, the drug vehicle control, equivalent to that present in the highest compound concentration was employed as a negative control. Results are expressed as mean ± SD. (**B**) Dose–response curve. A representative experiment out of 3 independent biological replicates.

**Figure 3 ijms-23-03254-f003:**
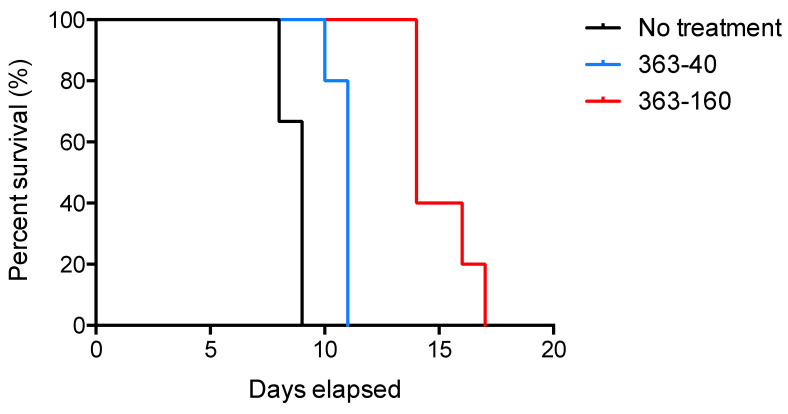
Survival curves of mice following acute toxoplasmosis. CD1 mice (*n* = 18) were infected with 100 RH tachyzoites and treated with JF363 at 40 mg/kg (*n* = 6) (363-40) or 160 mg/kg (*n* = 6) (363-160) or with PBS as a control (no treatment, *n* = 6) by oral route.

**Figure 4 ijms-23-03254-f004:**
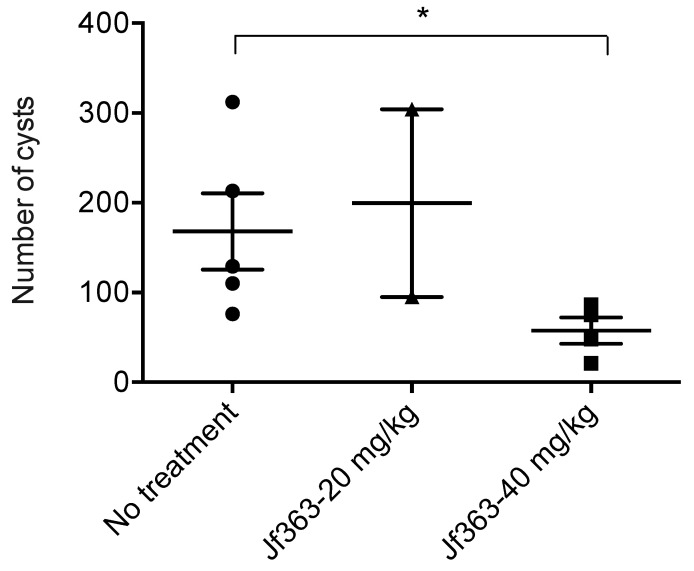
Cyst burden in infected mice. CBA mice were infected with 500 tachyzoites of type II Prugniaud strain. After 2 months, animals were not treated (*n* = 5) or treated with the JF363 drug at 20 mg/kg (*n* = 2) (363-20) or 40 mg/kg (*n* = 4) (363-40), via oral delivery. Cysts were counted 15 days later. Values were submitted to the Mann–Whitney test (* *p* < 0.05).

**Figure 5 ijms-23-03254-f005:**
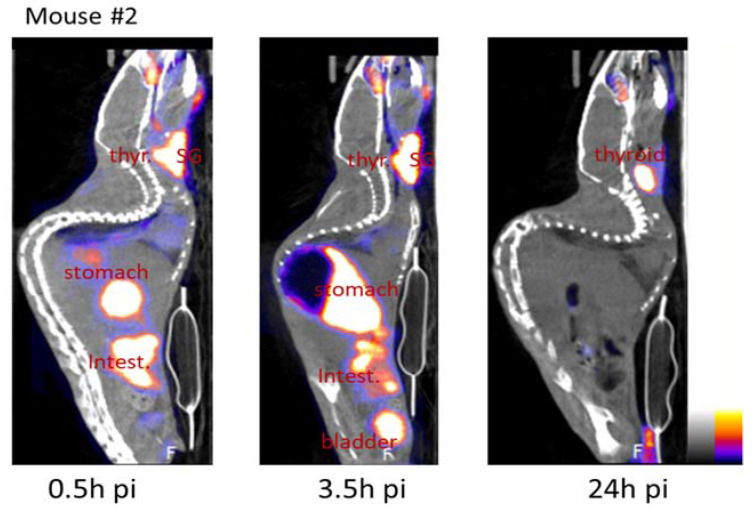
Examples of images collected from an infected mouse (sagittal view). Radioactivity was high in iodine-targeted tissues (stomach, intestine, salivary glands (SG)) and radiotracer entrance routes (stomach and intestine); the radiotracer was mainly eliminated through the urinary tract.

**Figure 6 ijms-23-03254-f006:**
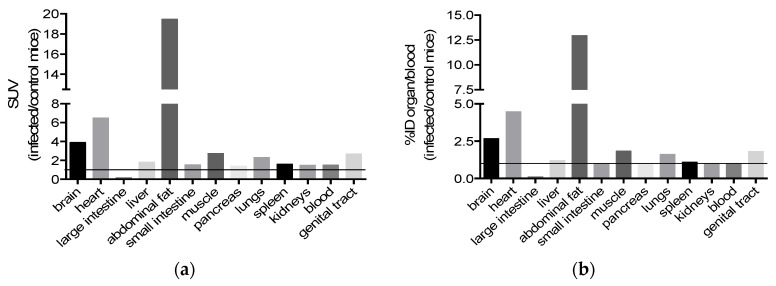
Biodistribution of the iodinated 363 molecules. The biodistribution ratios (infected/control mice) were determined following organ sampling at 24 h pi. Ratio are presented for (**a**) SUV = ((activity in organ/g of organ)/injected dose) × mouse body weight (g) and (**b**) the %ID/g of organ/%ID/g of blood.

**Figure 7 ijms-23-03254-f007:**
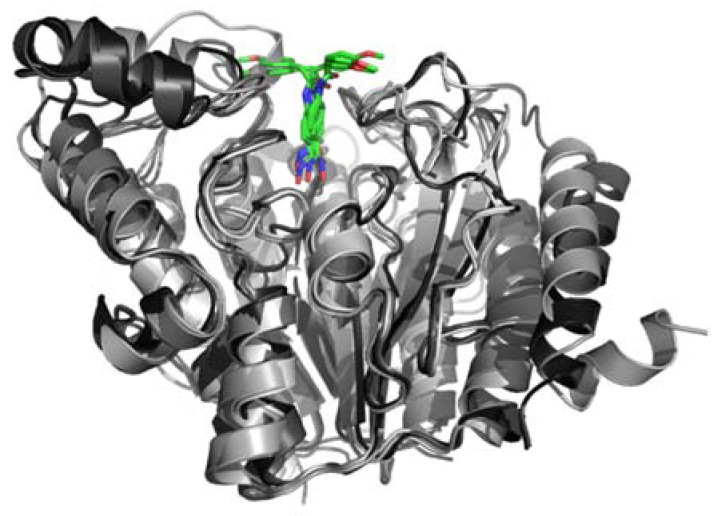
Models of Tg HDACs in complex with JF363. The best complexes for each of the five HDACs from *T. gondii* ME49 were superposed. The proteins are shown in grey ribbons and the ligand in colored sticks. The figure was drawn using Pymol (http://pymol.org, accessed on 20 February 2022).

**Figure 8 ijms-23-03254-f008:**
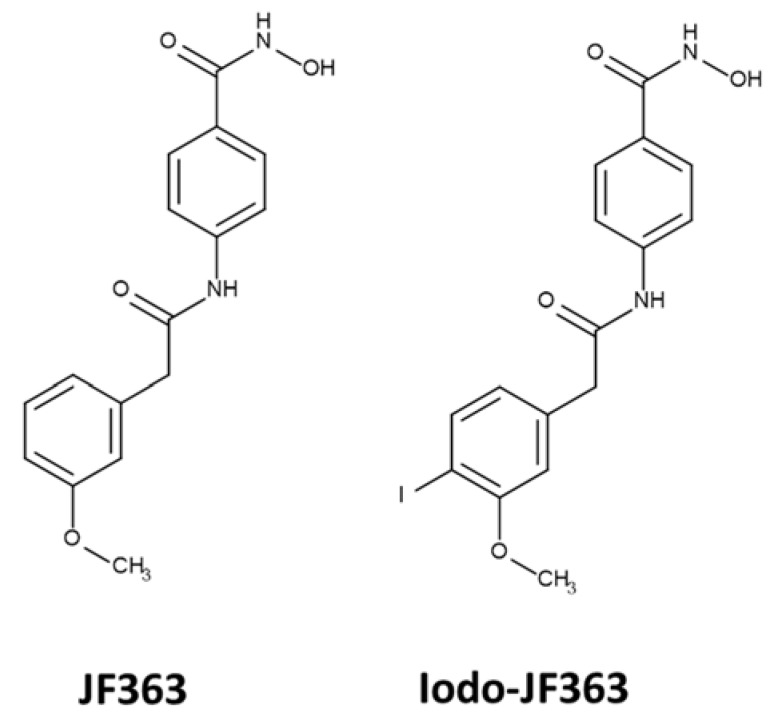
Chemical structure of the JF363 and iodo-JF363 used in the study.

**Table 1 ijms-23-03254-t001:** In vitro IC_50_ of JF363 ascertained against *Toxoplasma* with a wide range of genotypes and belonging to 7 of the 16 haplogroups described. IC_50_ data are the mean ± SD of 3 independent experiments.

Parasite Strain	Haplogroup	Clade	IC_50_ (μM)
RH-YFP	1	A	0.56 ± 0.05
Prugniaud	2	D	0.27 ± 0.02
VEG	3	C	0.18 ± 0.07
MAS	4	B	0.21 ± 0.01
GUY008-ABE	5	F	0.30 ± 0.21
GUY009-AKO	10	F	0.43 ± 0.12
GUY021-TOJ	10	F	0.27 ± 0.03
GUY-JAG1	11	/	0.17 ± 0.34

**Table 2 ijms-23-03254-t002:** Quantification of SPECT imaging in two infected mice. Results are expressed in %DI/cm^3^. pi: post-injection.

		Mouse #2			Mouse #3		
	**Time Pi**	0.5 h	3.5 h	24 h	0.5 h	3.5 h	24 h
**Organs**							
Stomach		12.1	17.1	4.2	45.0	42.7	8.9
Intestine		16.7	13.8	0.5	3.8	4.9	1.5
Salivary glands		16.7	37.6	3.0	8.1	20.3	3.3
Thyroid		28.9	18.8	26.6	4.1	9.9	29.5
Bladder		5.5	19.1	0.6	6.4	8.2	3.9
Heart		3.1	2.1	0.2	1.0	1.0	0.2
Lungs		1.8	1.5	0.1	0.6	0.5	0.2
Mouth		10.0	8.4	4.7	10.5	6.8	3.4
Neck ganglia		8.5	9.5	1.3	4.0	5.0	1.1

## Data Availability

All data generated or analysed during this study are included in this published article.

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
