# Peer review of "A Histone Deacetylase (HDAC) Inhibitor with Pleiotropic In Vitro Anti-Toxoplasma and Anti-Plasmodium Activities Controls Acute and Chronic Toxoplasma Infection in Mice"

_ijms, 2022, doi:10.3390/ijms23063254_

Round 1

Reviewer 1 Report

The authors investigated the in vitro antiparasitic effect of a specific inhibitor  (Jf363) of histone deacetylases from eight strains of Toxoplasma gondii. Anti-malarial activity of this inhibitor was observed for P. falciparum and P. berghei and it was found to be effective in inhibiting the proliferation of the parasites in the nanomolar range of concentrations.
In vivo anti-Toxoplasma effect was also studied and a significant effect was obtained at relatively low concentrations of the putative drug (20-40 mg/Kg in murine model).
JF363 seems to be excreted through the urinary tract of the mice, reaching 1% of the initial administered quantity in 24 h, and traces of the compound were identified in the brain, suggesting that the blood-brain barrier was crossed, a significant aspect of the study.
The authors presented some modeling/docking considerations, but this chapter is not fully developed for an in-depth review. 
The article is well structured in all chapters and the results are clearly presented. 
I was not able to identify details on the procedures followed in order to ensure ethics for in vivo studies.
The conclusions are sound.

Author Response

Thank you for having taking time to review our manuscript. Information about ethic in animal manipulation  is available in the section “institutional board statement” and ethics in animal

Reviewer 2 Report

The authors assessed in vitro anti-Toxoplasma and antimalarial (P. berghei hepatic stages and P. falciparum blood stages) activities of Jf363 (HDAC6 inhibitor). The experiments are well designed. More details can be added to the Methods section. The results are described clearly. The test compound seems to be quite promising as a potential drug candidate for Toxoplasma and/or malaria treatment.

The meaning of many of the abbreviations used in the text is not given. The authors should italicize consistently “in vitro” throughout the text. Toxoplasma (but not “toxoplasmosis”) and Plasmodium are Latin names that should be italicized. Subscripts and superscripts should be checked (IC50, O2, etc.).

Major comments:

1-Table 1: If several assays were performed for each strain, the authors should provide the information whether the IC50 values are means (or median) and add the standard deviation (or at least the range). IC50 (µM) (“50” subscript; Greek letter µ, instead of “u”); table title: Pleiotropic in vitro IC50 of Jf363 assessed in a large panel of T. gondii strains belonging to 7 of 16 known (?) haplogroups; I see 8 parasite strains in this table.

2-Table 2: The IC50 values given in this small table are also given in the main text and are illustrated in Figures 1 and 2. Table 2 is redundant. It should be deleted.

3-Similar text:

Lines 253-257: “Toxoplasma infection can be easily obtained in most laboratory animals. According to the strain used…histopathological examination.” These two sentences were copied from the English abstract in Derouin et al. [REF 38].

The text in some parts of the Methods also has high similarity with published papers. For example, lines 375-384 (Methods): “Each mouse brain was removed and homogenized… microscope Zeiss.” These lines were copied from Cavailles et al. PLoS Pathog 2014, 10(4):e1004005. Also some degree of similarity in lines 375-384 and 395-400 with Loeuillet et al. Int J Parasitol Drugs Drug Resist 2018, 8, 59-66.

These lines should be rephrased even if the original sources were the authors’ own papers (i.e., for Cavailles et al. and Loeuillet et al., but not for Derouin et al.).

Minor comments:

Introduction

Line 44: World Health Organization (WHO)

Line 45: The reference (WHO World Malaria Report 2021) should be cited as REF [1] and added to the list of references at the end of the manuscript. The web site can also be included in reference citation. Please check the instructions for authors.

Line 51: I would start a new paragraph from “Toxoplasmosis (not in italics) is an opportunistic infectious disease…”

Lines 55, 59: Toxoplasmosis (not in italics)

Line 58, 75: toxoplasmosis (small letter “t”)

Lines 64-65: Fansidar is pyrimethamine-sulfadoxine (not sulfadiazine). Please check.

Line 66, 84, 86: tachyzoites (no need for two dots over “i” as in French)

Line 72: human immunodeficiency virus (HIV)

Line 80: IC50 (50, subscript)

Line 88: delete the comma after (HDACs); antiplasmodial (spelling)

Line 92-94: we characterized…with high selectivity and (?) high anti-HDAC6… in vitro (in italics)

Methods:

Lines 291-292: B. Striepen (Athens, Greece)… DJ Bzik (Lebanon, USA)…

Line 301: CO2 (“2” subscript)

Lines 304-305: Instituto de Medicina Molecular (Lisbon, Portugal)

Line 305: Please add a short description of the P. falciparum strain used (NF54), how this strain was obtained, etc.

Line 309: “1300 rpm” Please express it in terms of x g.

Line 311: phosphate-buffered saline (PBS)

Line 312: “various concentrations ranging from 1 to 2 µM”: Please specify whether each concentration was tested in duplicate (or triplicate) in each assay. The authors can also add here that 3 independent assays were performed (as stated in line 126-127).

Line 313: 5 (instead of “five”) µg/mL

Line 318: bovine serum albumin (BSA)

Line 323: parasitic index (PI)…

Line 327: (50, subscript)

Line 328: delete “December 2008” and add instead (San Diego, CA)

Line 332: N-2-hydroxyethylpiperazine-N’-2-ethanesulfonic acid (HEPES)

Line 334: In vitro, Plasmodium (in italics)

Line 335: Liver (capital letter “L”)

Line 336: “as previously described” Please add the appropriate reference.

Line 338: dimethyl sulfoxide (DMSO)

Line 340: “test concentrations” Please provide the final concentrations.

Line 342: 1800 x g

Lines 344, 346: Please provide the city and country of these suppliers.

Line 348: The software developer’s address should have been given in line 328. Delete “December 2008.”

Lines 350-351: “cultures of P. falciparum NF54 strain…were incubated”

Lines 352-353: The concentrations of the stock solution and working solutions should be given. The authors should also give the range of final concentrations used in their assays.

Line 354: consisted

Line 356: Albumax (supplier?; city?, country?); NaHCO3 (“3” subscript); Five (not “5”) microliters…

Line 362: “(Prism 4 for MacOSX, …December 2008)” can be deleted.

Line 364: Janvier Labs (Le Genest-Saint-Isle, France ??)

Line 365-366: once a day

Line 367: “…were treated with the vehicle alone”

Line 372: A space after “previously”

Line 374: the number of cysts in the brain was counted/evaluated

Line 377: ethylenediaminetetraacetic acid [EDTA] 1 mM, sodium dodecycl sulfate (SDS) 0.2%

Line 378: phenylmethylsulfonyl fluoride (PMSF)

Line 380: fluorescein isothiocyanate (FITC)

Line 386: in figure 7

Line 389: Chemical structure of Jf363… (delete “the”)

Lines 392-393: DCM, DIEA, HOBt (abbreviations), thin layer chromatography (TLC)

Line 422: TLC (this abbreviation should have been introduced in lines 392-393)

Line 427: T. gondii (in italics)

Line 430: O2 (“2” subscript); single-photon emission computerized tomography/computed tomography (SPECT/CT)

Line 433: Mediso (city?, country?)

Line 434: cm3 (“3” in superscript)

Line 436: %DI/cm3, Should it be %ID (for injected dose)? See line 444 (% ID/g).

Lines 439-441: organs and tissue samples or biological samples?  Is urine an organ?  

Line 442: Is there a word missing here? A gamma counter?

Lines 444-445: the bodyweight was different

Lines 449-450: the proportion…was calculated

Line 451: Because T. gondii…

Line 453: film which was then scanned

Line 453: …uptake as % ID/g (this abbreviation was already given earlier in the text).

Line 458: T. gondii (spelling)

Lines 462-463: better matched…and allowed the modeling…

Results:

Lines 103 and 113: Toxoplasma, Plasmodium in italics

Line 105: concentrations ranging from 0 to 2 µM; tested drug concentrations, 1 nM, 10 nM, 25 nM, 50 nM, 100 nM, 250 nM, 500 nM ??)

Line 115: rodent malaria P. berghei…human malaria P. falciparum

Lines 126 and 141-142: Representative experiment out of 3 independent biological replicates.

Line 132: IC50

Line 135-136: in vitro (in italics)

Line 141: results are expressed as…

Line 146: T. gondii (in italics)

Line 148: we analyzed

Line 150: mice…survived, and they did not present

Line 151: all untreated…

Line 153: p < 0.0001

Line 152: Figure 3??

Line 159: The figure legend should explain what “363-40” and “363-160” mean, even if it may be quite obvious: Jf363 at 40 mg/kg and Jf363 at 160 mg/kg.

Line 160: cyst formation in chronically infected mice

Line 161: Toxoplasma (italics) cyst formation

Line 163: many cysts (mean ± SD, range; 168 ± ??, 76 – 312) were detected

Line 165: a significantly (P < 0.05) lower number of cysts (mean ± SD, range; 57 ± ??, 21 – 85) were detected

Line 166: Figure 4??

Fig. 4: Y-axis legend – number of cysts

Lines 174-175: “As already reported,…” The sentence should be written for more clarity. The sentence structure, in particular the subject of the sentence, is not clear (deionization? radioactivity?).

Lines 179-181: …imaging at 30 min post-injection (pi); ID/cm3 (superscript)

Line 180: to a lesser extent

Table 3. Why is “mouth” between quotation marks?

Lines 188 and 200: What does “SUV” stand for?

Line 194: an increase in the presence of Jf363…

Line 199-200: Ratio is OR Ratios are

Lines 209-216: The results should be generally in the past tense. “docking…is/was obtained” “this screening predicts/predicted…” 

Discussion:

Lines 224, 226, 227: we confirmed, we investigated… proved ?? its efficiency in controlling Toxoplasma (italics)

Line 230: varied

Line 231: fell within the range

Line 233: against Toxoplasma strains with different genotypes

Line 245: against Toxoplasma with a wide range of genotypes

Line 248: against various strains of T. gondii (spelling) that we tested. Further structural analysis would help to derive more potent…

Line 249: antiparasitic drug that is potent against parasites…reinforces…

Line 266: increasing the drug dose to 200 or 250 mg/kg may improve…

Line 273: compared to untreated animals

Line 274-275: The majority…have been tested…

Line 278: inhibitors hold promise as a …

Line 279: its (spelling) in vivo efficiency

Line 284: tachyzoites

Conclusion:

Lines 468-469: “against several Toxoplasma genotypes” The drug is active against Toxoplasma strains with different genotypes. It is not active against a genotype.

Line 471: can be easily synthesized

Line 473: in vitro (in italics)

Lines 473-474: The meaning of the sentence (“toxicity challenge is several cell lines [24], and we overlook toxicity in the treated mice”) is not very clear. Please clarify this sentence and close the parenthesis where necessary.

References:

REF: The references should have the same format. The article title is not capitalized, except for the first letter of the first word and proper names.

REF 1: falciparum (small letter “f”)

REF 16: Clin Microbiol Rev 2018, 31, e00057-17.

REF 24, 25, 26, 27: The journal names should be in the abbreviated form: Int J Parasitol Drugs Drug Resist; Int J Mol Sci; J Parasitol; Proc Natl Acad Sci USA.

REF 25: Int J Mol Sci 2019, 20, 2973. doi: 10.3390/ijms20122973.

REF 41: Front Microbiol 2017, 8, 25.

Author Response

Reviewer 2

The authors assessed in vitro anti-Toxoplasma and antimalarial (P. berghei hepatic stages and P. falciparum blood stages) activities of Jf363 (HDAC6 inhibitor). The experiments are well designed. More details can be added to the Methods section. The results are described clearly. The test compound seems to be quite promising as a potential drug candidate for Toxoplasma and/or malaria treatment.

The meaning of many of the abbreviations used in the text is not given. The authors should italicize consistently “in vitro” throughout the text. Toxoplasma (but not “toxoplasmosis”) and Plasmodium are Latin names that should be italicized. Subscripts and superscripts should be checked (IC50, O2, etc.).

Thank you for the carefully reviewing of our manuscripts and for the time that you have spent on this manuscript. Thank you for the remarks these corrections were performed in the new version.

Major comments:

1-Table 1: If several assays were performed for each strain, the authors should provide the information whether the IC50 values are means (or median) and add the standard deviation (or at least the range). IC50 (µM) (“50” subscript; Greek letter µ, instead of “u”); table title: Pleiotropic in vitro IC50 of Jf363 assessed in a large panel of T. gondii strains belonging to 7 of 16 known (?) haplogroups; I see 8 parasite strains in this table.

We have added the information on the table legend.

2-Table 2: The IC50 values given in this small table are also given in the main text and are illustrated in Figures 1 and 2. Table 2 is redundant. It should be deleted.

 The table is deleted in the new version.

3-Similar text:

Lines 253-257: “Toxoplasma infection can be easily obtained in most laboratory animals. According to the strain used…histopathological examination.” These two sentences were copied from the English abstract in Derouin et al. [REF 38].

The text in some parts of the Methods also has high similarity with published papers. For example, lines 375-384 (Methods): “Each mouse brain was removed and homogenized… microscope Zeiss.” These lines were copied from Cavailles et al. PLoS Pathog 2014, 10(4):e1004005. Also some degree of similarity in lines 375-384 and 395-400 with Loeuillet et al. Int J Parasitol Drugs Drug Resist 2018, 8, 59-66.

These lines should be rephrased even if the original sources were the authors’ own papers (i.e., for Cavailles et al. and Loeuillet et al., but not for Derouin et al.).

All these sentences are rephrased in the new version.

Minor comments:

Introduction

Line 44: World Health Organization (WHO) corrected

Line 45: The reference (WHO World Malaria Report 2021) should be cited as REF [1] and added to the list of references at the end of the manuscript. The web site can also be included in reference citation. Please check the instructions for authors. Reference added

Line 51: I would start a new paragraph from “Toxoplasmosis (not in italics) is an opportunistic infectious disease…” Done

Lines 55, 59: Toxoplasmosis (not in italics) Corrected

Line 58, 75: toxoplasmosis (small letter “t”) Corrected

Lines 64-65: Fansidar is pyrimethamine-sulfadoxine (not sulfadiazine). Please check. Corrected

Line 66, 84, 86: tachyzoites (no need for two dots over “i” as in French) Corrected

Line 72: human immunodeficiency virus (HIV) Corrected

Line 80: IC50 (50, subscript) Corrected

Line 88: delete the comma after (HDACs); antiplasmodial (spelling) Corrected

Line 92-94: we characterized…with high selectivity and (?) high anti-HDAC6… in vitro (in italics) Corrected

Methods:

Lines 291-292: B. Striepen (Athens, Greece)… DJ Bzik (Lebanon, USA)… Corrected

Line 301: CO2 (“2” subscript) Corrected

Lines 304-305: Instituto de Medicina Molecular (Lisbon, Portugal) Corrected

Line 305: Please add a short description of the P. falciparum strain used (NF54), how this strain was obtained, etc. Corrected

Line 309: “1300 rpm” Please express it in terms of x g. Corrected

Line 311: phosphate-buffered saline (PBS) Corrected

Line 312: “various concentrations ranging from 1 to 2 µM”: Please specify whether each concentration was tested in duplicate (or triplicate) in each assay. The authors can also add here that 3 independent assays were performed (as stated in line 126-127). Corrected

Line 313: 5 (instead of “five”) µg/mL Corrected

Line 318: bovine serum albumin (BSA) Corrected

Line 323: parasitic index (PI)… Corrected

Line 327: (50, subscript) Corrected

Line 328: delete “December 2008” and add instead (San Diego, CA). Corrected

Line 332: N-2-hydroxyethylpiperazine-N’-2-ethanesulfonic acid (HEPES). Corrected

Line 334: In vitro, Plasmodium (in italics)

Line 335: Liver (capital letter “L”). Done

Line 336: “as previously described” Please add the appropriate reference. Done

Line 338: dimethyl sulfoxide (DMSO). Done

Line 340: “test concentrations” Please provide the final concentrations. Done

Line 342: 1800 x g Done

Lines 344, 346: Please provide the city and country of these suppliers. Done

Line 348: The software developer’s address should have been given in line 328. Delete “December 2008.” Done

Lines 350-351: “cultures of P. falciparum NF54 strain…were incubated” Done

Lines 352-353: The concentrations of the stock solution and working solutions should be given. The authors should also give the range of final concentrations used in their assays. Done

Line 354: consisted, corrected

Line 356: Albumax (supplier?; city?, country?); NaHCO3 (“3” subscript); Five (not “5”) microliters…Corrected

Line 362: “(Prism 4 for MacOSX, …December 2008)” can be deleted. Corrected

Line 364: Janvier Labs (Le Genest-Saint-Isle, France ??) yes true corrected

Line 365-366: once a day corrected

Line 367: “…were treated with the vehicle alone” Corrected

Line 372: A space after “previously”  added thank you

Line 374: the number of cysts in the brain was counted/evaluated, counted thank you for the remark

Line 377: ethylenediaminetetraacetic acid [EDTA] 1 mM, sodium dodecycl sulfate (SDS) 0.2% added

Line 378: phenylmethylsulfonyl fluoride (PMSF) added

Line 380: fluorescein isothiocyanate (FITC) added

Line 386: in figure 7 added

Line 389: Chemical structure of Jf363… (delete “the”) deleted

Lines 392-393: DCM, DIEA, HOBt (abbreviations), thin layer chromatography (TLC) added

Line 422: TLC (this abbreviation should have been introduced in lines 392-393) added

Line 427: T. gondii (in italics)  Corrected

Line 430: O2 (“2” subscript); single-photon emission computerized tomography/computed tomography (SPECT/CT) Corrected

Line 433: Mediso (city?, country?) added

Line 434: cm3 (“3” in superscript) corrected

Line 436: %DI/cm3, Should it be %ID (for injected dose)? See line 444 (% ID/g). Corrected

Lines 439-441: organs and tissue samples or biological samples?  Is urine an organ?   Thank you for the remark the correction is performed.

Line 442: Is there a word missing here? A gamma counter? Yes it is a gamma counter, correction performed

Lines 444-445: the bodyweight was different. Grammatical error corrected

Lines 449-450: the proportion…was calculated. Grammatical error corrected

Line 451: Because T. gondii… Corrected

Line 453: film which was then scanned. Corrected

Line 453: …uptake as % ID/g (this abbreviation was already given earlier in the text). Corrected

Line 458: T. gondii (spelling) corrected

Lines 462-463: better matched…and allowed the modeling… Corrected

Results:

Lines 103 and 113: Toxoplasma, Plasmodium in italics. Corrected

Line 105: concentrations ranging from 0 to 2 µM; tested drug concentrations, 1 nM, 10 nM, 25 nM, 50 nM, 100 nM, 250 nM, 500 nM ??) yes for T. gondii 0 to 2 mM

Line 115: rodent malaria P. berghei…human malaria P. falciparum corrected

Lines 126 and 141-142: Representative experiment out of 3 independent biological replicates. Corrected

Line 132: IC50. Corrected

Line 135-136: in vitro (in italics). Corrected

Line 141: results are expressed as… Corrected

Line 146: T. gondii (in italics) Corrected

Line 148: we analyzed. Corrected

Line 150: mice…survived, and they did not present. Corrected

Line 151: all untreated… Corrected

Line 153: p < 0.0001 Corrected

Line 152: Figure 3?? Corrected

Line 159: The figure legend should explain what “363-40” and “363-160” mean, even if it may be quite obvious: Jf363 at 40 mg/kg and Jf363 at 160 mg/kg. Information added in the figure’s legend.

Line 160: cyst formation in chronically infected mice Corrected

Line 161: Toxoplasma (italics) cyst formation. Error corrected

Line 163: many cysts (mean ± SD, range; 168 ± ??, 76 – 312) were detected. Changes were performed

Line 165: a significantly (P < 0.05) lower number of cysts (mean ± SD, range; 57 ± ??, 21 – 85) were detected. Corrected

Line 166: Figure 4?? Sorry for this confusion it is effectively figure 4

Fig. 4: Y-axis legend – number of cysts

Lines 174-175: “As already reported,…” The sentence should be written for more clarity. The sentence structure, in particular the subject of the sentence, is not clear (deionization? radioactivity?). Sentence rephrased

Lines 179-181: …imaging at 30 min post-injection (pi); ID/cm3 (superscript). Corrected

Line 180: to a lesser extent Corrected

Table 3. Why is “mouth” between quotation marks?

Lines 188 and 200: What does “SUV” stand for? corrected

Line 194: an increase in the presence of Jf363…Corrected

Line 199-200: Ratio is OR Ratios are. Corrected

Lines 209-216: The results should be generally in the past tense. “docking…is/was obtained” “this screening predicts/predicted…”  Corrected,

Discussion:

Lines 224, 226, 227: we confirmed, we investigated… proved ?? its efficiency in controlling Toxoplasma (italics) Corrected

Line 230: varied Corrected

Line 231: fell within the range. Corrected

Line 233: against Toxoplasma strains with different genotypes Corrected

Line 245: against Toxoplasma with a wide range of genotypes Corrected

Line 248: against various strains of T. gondii (spelling) that we tested. Further structural analysis would help to derive more potent… Corrected

Line 249: antiparasitic drug that is potent against parasites…reinforces… Corrected

Line 266: increasing the drug dose to 200 or 250 mg/kg may improve…

Line 273: compared to untreated animals Corrected

Line 274-275: The majority…have been tested… Corrected

Line 278: inhibitors hold promise as a …Corrected

Line 279: its (spelling) in vivo efficiency Corrected

Line 284: tachyzoites Corrected

Conclusion:

Lines 468-469: “against several Toxoplasma genotypes” The drug is active against Toxoplasma strains with different genotypes. It is not active against a genotype. Corrected

Line 471: can be easily synthesized Corrected

Line 473: in vitro (in italics) Corrected

Lines 473-474: The meaning of the sentence (“toxicity challenge is several cell lines [24], and we overlook toxicity in the treated mice”) is not very clear. Please clarify this sentence and close the parenthesis where necessary. Corrected

References:

REF: The references should have the same format. The article title is not capitalized, except for the first letter of the first word and proper names. Corrected

REF 1: falciparum (small letter “f”) Corrected

REF 16: Clin Microbiol Rev 2018, 31, e00057-17. Corrected

REF 24, 25, 26, 27: The journal names should be in the abbreviated form: Int J Parasitol Drugs Drug Resist; Int J Mol Sci; J Parasitol; Proc Natl Acad Sci USA. Corrected

REF 25: Int J Mol Sci 2019, 20, 2973. doi: 10.3390/ijms20122973. Corrected

REF 41: Front Microbiol 2017, 8, 25. Corrected
